# Regulation of Phosphoinositide Signaling by Scaffolds at Cytoplasmic Membranes

**DOI:** 10.3390/biom13091297

**Published:** 2023-08-24

**Authors:** Tianmu Wen, Narendra Thapa, Vincent L. Cryns, Richard A. Anderson

**Affiliations:** 1School of Medicine and Public Health, University of Wisconsin-Madison, 1111 Highland Avenue, Madison, WI 53705, USA; twen4@wisc.edu (T.W.); nthapa@wisc.edu (N.T.); 2Department of Medicine, University of Wisconsin Carbone Cancer Center, School of Medicine and Public Health, University of Wisconsin-Madison, 1111 Highland Avenue, Madison, WI 53705, USA

**Keywords:** phosphoinositide, scaffolding protein, PI3K-Akt pathway, autophagy, lysosome

## Abstract

Cytoplasmic phosphoinositides (**PI**) are critical regulators of the membrane–cytosol interface that control a myriad of cellular functions despite their low abundance among phospholipids. The metabolic cycle that generates different PI species is crucial to their regulatory role, controlling membrane dynamics, vesicular trafficking, signal transduction, and other key cellular events. The synthesis of phosphatidylinositol (3,4,5)-triphosphate (PI3,4,5P_3_) in the cytoplamic PI3K/Akt pathway is central to the life and death of a cell. This review will focus on the emerging evidence that scaffold proteins regulate the PI3K/Akt pathway in distinct membrane structures in response to diverse stimuli, challenging the belief that the plasma membrane is the predominant site for PI3k/Akt signaling. In addition, we will discuss how PIs regulate the recruitment of specific scaffolding complexes to membrane structures to coordinate vesicle formation, fusion, and reformation during autophagy as well as a novel lysosome repair pathway.

## 1. Introduction

Phosphoinositides (**PI**) are a family of minor phospholipids originating from phosphatidylinositol. They possess a glycerol backbone that is linked largely to arachidonic acid and stearic acid and to the inositol head through a phosphodiester bond. PIs are amphiphilic; while localized in membrane structures, their polar inositol group faces the cytoplasm and their non-polar hydrophobic fatty acid tails are embedded in the lipid bilayer. The 3, 4, and 5 hydroxyls on the inositol head can be further phosphorylated in all combinations generating a total of seven phosphorylation variants of the PI family [1,2]. These can interconverted by a large array of PI kinases and phosphatases [1]. PIs exist on the cytoplasmic face of all cell membrane structures and interact with many membrane proteins to regulate cell motility, membrane trafficking, nuclear events, and the generation of PI-derived messengers via receptor signaling [3]. PIs regulate a broad array of cellular functions in eukaryotes. 

For the generation of PI messengers, it is vital that enzymes in these pathways are regulated efficiently and with high fidelity in a temporal and spatial mode [4]. To accomplish this, there is emerging evidence that scaffold proteins act as platforms on which components of the pathways assemble to lower entropy and enhance specificity [5]. Scaffold proteins facilitate signaling through multiple mechanisms, providing proximity, localization, or inhibiting interactions. The functional role of scaffold proteins for protein kinase and similar pathways are well defined [6,7]. Recent studies have implicated several scaffold proteins in PI signaling, which is the focus of this review and commentary. 

## 2. Cytoplasmic Phosphoinositides and Metabolism

There are eight isomers of the PI family, including the unphosphorylated phosphatidylinositol (**PtdIns**); three singly phosphorylated species: PtdIns 3-phosphate (**PI(3)P**), PtdIns 4-phosphate (**PI(4)P**), and PtdIns 5-phosphate (**PI(5)P**); three doubly phosphorylated species: PtdIns (3,4)-bisphosphate (**PI(3,4)P_2_**), PtdIns (3,5)-bisphosphate (**PI(3,4)P_2_**), and PtdIns (4,5)-bisphosphate (**PI(4,5)P_2_**); and the triply phosphorylated isomer PtdIns (3,4,5)-triphosphate (**PI(3,4,5)P_3_**). PtdIns is the precursor of all PIs, and the most abundant among cellular inositol lipids, accounting for over 90% of all PI species [8]. PI(4)P and PI(4,5)P_2_ are the most abundant of the singly and doubly phosphorylated species, each making up ~2–5% of total PIs and constituting approximately 90% of phosphorylated PIs [8,9,10,11]. The two other monophosphates, PI(3)P and PI(5)P, constitute ~0.2–0.5% and ~0.01–0.2% of total cellular PIs, respectively [8,12]. The other PI species, PI(3,4)P_2_, PI(3,5)P_2_, and PI(3,4,5)P_3_, are expressed at lower concentrations in resting cells, and are often undetectable without stimulation [8,9,11].

The PI signaling cycle was first discovered by Lowell and Mable Hokin in 1953 [13] and the molecular details were uncovered over subsequent decades. (Figure 1) PtdIns is synthesized via the conjugation of cytidine diphosphate diacylglycerol (**CDP-DAG**) with myo-inositol in the endoplasmic reticulum (**ER**) by a PI synthesis enzyme (**PIS**) [8,14]. PtdIns is transported from the ER via either traditional vesicular transport or lipid transfer proteins, including TMEM24 [15], phosphoinositide transfer proteins (PITPs) [16], Nir2, and Nir3 [17,18,19,20], at membrane contact sites (**MCS**).

PI(3)P is generated via (1) the phosphorylation of the 3-position hydroxyl of PtdIns by class II PtdIns 3-kinases (**PI3K-C2**) and class III PtdIns 3-kinase Vps34 [21,22,23], or (2) the dephosphorylation of PI(3,4)P_2_ by INPP4A and INPP4B [24,25,26]. PI(3)P plays an important role in membrane dynamics and trafficking and is most abundant in early endosome compartments. PI(3)P is also dephosphorylated to PtdIns by myotubularin family phosphatases [8].

PI(4)P is produced via the phosphorylation of the 4-position hydroxyl of PtdIns by PtdIns 4-kinases (**PI4Ks**) or via the dephosphorylation of PI(3,4)P_2_ and PI(4,5)P_2_ by the respective phosphatases [8]. PI(4)P is localized on several subcellular compartments, including the plasma membrane and the Golgi, particularly the trans-Golgi. The PI(4)P population on the plasma membrane is mostly generated by PI4KIIIα [27], while the PI(4)P population on the Golgi is generated by PI4KIIIβ, PI4KIIα, and PI4KIIβ [28,29,30,31]. Golgi PI(4)P is converted back into PtdIns by a PI(4)P 4-phosphatase, Sac1 [32].

PI(5)P is synthesized via the phosphorylation of the 5-position hydroxyl of PtdIns by PtdIns 5-kinase PIKfyve [33], from PI(3,5)P_2_ by myotubularin phosphatases [34], or from PI(4,5)P_2_ by type I and type II PI(4,5)P_2_ 4-phosphatases [35,36,37]. PI(5)P is found on the plasma membrane and endomembranes, participating in Akt/mTOR signaling [38], apoptosis [39], and nuclear stress signal transduction [40].

PI(3,4)P_2_ can be generated via the phosphorylation of the 3-position hydroxyl of PI(4)P by PI3K-C2 on the plasma membrane [41]; however, it was shown that this is not the major synthesis pathway [42]. The majority of PI(3,4)P_2_ is produced from PI(3,4,5)P_3_ by SHIP1 (also known as INPP5D) and SHIP2 (also known as INPPL1) phosphatases [43,44]. PI(3,4)P_2_ localizes on the plasma membrane and endocytic compartments [8] and is dephosphorylated by INPP4A/B to PI(3)P or by PTEN to PI(4)P [45,46]. 

PI(3,5)P_2_ is synthesized via the phosphorylation of the 5-position hydroxyl on PI(3)P by PIKfyve [47,48]. PI(3,5)P_2_ is the signature PI on late endosomes, inducing the release of cortactin from the endosomal branched actin network, regulating endosomal fission and fusion, which are crucial for vesicle trafficking [49].

PI(4,5)P_2_ is predominantly generated via the phosphorylation of the 5-position hydroxyl of PI(4)P by the three isoforms of PI(4)P 5-kinases (**PIPKIα**, **PIPKIβ**, and **PIPKIγ**) [50,51]. The PIPKIβ homo-dimer and the PIPKIβ- PIPKIγ hetero-dimer are thought to be important for enzymatic activity and localization to the plasma membrane [52]. In addition, a minor population of PI(4,5)P_2_ is generated from PI(5)P by the type II PI(5)P 4-kinases (**PIPKIIα**, **PIPKIIβ**, and **PIPKIIγ**). PI(4,5)P_2_ is also produced via the dephosphorylation of PI(3,4,5)P_3_ by phosphatases such as PTEN and TPIPα/β/γ [11]. The major constitutive pool of PI(4,5)P_2_ is localized on the plasma membrane, where it is regulated by two mechanisms: hydrolysis by **phospholipase C** (PLC) to produce inositol 1,4,5-triphosphate (**IP_3_**) and DAG in a receptor-mediated manner, or dephosphorylation by various PI(4,5)P_2_ 5-phosphatases, including SYNJ1, SYNJ2, OCRL, INPP5B, INPP5E, INPP5J, and INPP5K, to produce PI(4)P [8].

PI(3,4,5)P_3_ in the cytoplasm is synthesized from PI(4,5)P_2_ by the isoforms of the class I PtdIns 3-kinase (**PI3K**) [53]. The class I PI3Ks are heterodimers composed of one adaptor subunit and one catalytic subunit. Based on their subunits and mediators, they can be further divided into two subclasses: IA and IB. Class IA PI3Ks are activated by receptor tyrosine kinases, and are composed of one of the three catalytic subunits, p110α, p110β, or p110δ, together with one of the five adaptor subunits, p85α, p85β, p55α, p55γ, and p50α. Class IB PI3Ks are activated by G-protein-coupled receptors, with only one catalytic subunit variant, p110γ, and two adaptor subunit variants, p101 and p84 [54]. Although PI(3,4,5)P_3_ is barely detectable in quiescent cells, its cellular concentration increases rapidly, up to 100-fold within seconds in response to diverse stimuli [55,56]. PI(3,4,5)P_3_ is central to multiple key signaling pathways controlling proliferation, migration, growth, metabolism and apoptosis. One of these pathways is the PI3K/Akt/mTOR pathway, where the PI3,4,5P_3_ generated by PI3K recruits Akt and the Akt-activating kinases PDK1 and mTORC2 via their pleckstrin homology (PH) domain to membranes to phosphorylate Akt on Thr308 and Ser473, activating Akt. Active Akt in turn phosphorylates and inhibits the mTOR complex 1 (**mTORC1**) inhibitors TSC1/2 [57]. Both Akt and mTORC1 are master regulators of cellular activity and oncogenic events. The major regulator of PI3,4,5P_3_ is PTEN, which dephosphorylates it back to PI(4,5)P_2_ [11]. A loss of PTEN and activated missense mutations of the PI3K catalytic subunit gene are often found in many cancers, resulting in hyperactivated PI3K signaling and dysregulated growth [58,59].

## 3. Scaffolding Complexes Target the PI3K-Akt Pathways to Specific Compartments

### 3.1. The CNK1 Scaffolding Complex Activates PIPKIs upon Insulin Stimulation

The connecter enhancer of KSR1 (**CNK1**) is a member of the mammalian CNK family, which possesses a sterile α motif (**SAM**), a conserved region in the CNK (**CRIC**) domain, a PSD-95/DLG-1/ZO-1 (**PDZ**) domain which binds to the C-terminus of proteins, a **PH** domain recognizing PIs, and a cytohensin-binding domain, indicating its potential as a scaffolding protein [60]. In 2010, Lim et al. identified the C-terminal binding domain of CNK1 constitutively coupled with the coiled-coil domain of the cytohensins, recruiting cytohensin-2 to the plasma membrane in an insulin stimulation-specific manner [61]. The CNK1/cytohensin complex further recruits allosteric activators of PIPKIs, Arf6 and Arf1, to the site of stimulation, enhancing PI(4,5)P_2_ synthesis to enable PI3K/Akt signaling upon insulin stimulation (Figure 2).

CNK1 is phosphorylated by Akt at Ser22 in the SAM region to induce its oligomerization and increase Akt binding, while phosphorylation at Thr8 in the SAM region, stimulated by EGF, counteracts the positive feedback [62]. 

### 3.2. EBP50 Inhibits PI3K/Akt Signaling by Recruiting PTEN to the Membrane

Ezrin-radixin-moesin-binding phosphoprotein 50 (**EBP50**), also known as Na^+^/H^+^ exchanger regulatory factor 1 (**NHERF1**), is a scaffolding protein containing two PDZ domains [63]. The PDZ-1 domain of EBP50 binds PTEN and the transmembrane platelet-derived growth factor receptor (**PDGFR**) [64], while the PDZ-2 domain of EBP50 binds Akt [65], localizing PTEN to membrane Akt activation sites in a PDGF-stimulated manner. EBP50 forms dimers in cells to stabilize activated PTEN, facilitating the dephosphorylation of PI(3,4,5)P_3_ to PI(4,5)P_2_ to inhibit PI3K/Akt signaling [66] (Figure 2). EBP50 also recruits the Akt phosphatase PHLPP1 to the plasma membrane, disrupting PI3K/Akt signaling [66]. Thus, EBP50 acts as a tumor suppressor when localized on membranes; however, in glioblastoma cells, EBP50 translocates from the membrane to the cytoplasm, bringing PTEN with it, thereby activating PI3K/Akt signaling at membranes [67].

### 3.3. PI3,4,5P3-Bound PHB Scaffolds SHIP1 to Inhibit Akt Activation

Prohibitin (**PHB**) is a highly conserved, ubiquitously expressed protein that is targeted toward the nucleus, cytoplasm, and mitochondria where it has site-specific functions [68]. There are two variants of PHB, PHB1 and PHB2, which form heterodimers that interact with PI(3,4,5)P_3_ and Akt [69,70]. The insulin receptor directly binds and phosphorylates PHB1 at Tyr114 upon insulin stimulation, promoting the recruitment of the PI(3,4,5)P_3_ 5-phosphatase SHIP1 by PHB1 to sites of PI3K/Akt activation, resulting in the inhibition of the insulin-induced PI3K/Akt pathway [7]. (Figure 2) Interestingly, constitutively activated Akt can phosphorylate PHB2 at Thr258 to disrupt PHB-PI(3,4,5)P_3_ and PHB-SHIP1 binding, promoting Akt activation as positive feedback [69].

### 3.4. DAB2IP Blocks p85 and Akt Localization to the Membranes 

The deletion of ovarian carcinoma 2/disabled homolog 2-interacting protein (**DAB2IP**), also known as ASK1-interacting protein-1 (**AIP1**) is a member of the Ras-GAP (GTPase-activating protein) family [71]. DAB2IP interacts with the PI3K adaptor subunit p85 through its proline-rich domain and Akt through its PERIOD-like domain, forming a scaffolding complex. With TNF-α stimulation, the complex translocates to the cytosol, blocking the membrane recruitment of p85 and Akt and inhibiting PI3K/Akt signaling. (Figure 2) These interactions are enhanced via the Ser604 phosphorylation of DAB2IP by ASK1 and disrupted by the phosphorylation of DAB2IP at Ser847 [72].

### 3.5. APPL1 Scaffolds Akt2 on Endosomes for PI(3,4)P_2_ Activation

The adaptor protein containing PH domain, PTB domain, and Leucine zipper motif-1 (**APPL1**) consists of multiple domains and motifs for protein and signaling molecule interaction, as its name suggests [73]. In response to growth factor stimulation, PI(3,4)P_2_ is generated on endosomes by the phosphatase SHIP2 from PI(3,4,5)P_3_ [42]. APPL1 interacts with PI(3,4)P_2_ on endosomes [42,74], and facilitates Akt2 activation via insulin [75], NGF-1 [76], and androgen stimulation [77]. With stimulation, PI3K-C2γ is recruited to early endosomes by Rab-GTP and produces PI(3,4)P_2_, which activates the APPL1-Akt2 complex recruited to early endosomes by Rab5 [78] (Figure 2).

### 3.6. PI(4,5)P_2_ Mediated LAPTM4B Regulates EGFR Sorting and Degradation

LAPTM4B is a member of the mammalian lysosomal-associated protein transmembrane family that localizes to the late endosome/lysosome [79]. It possesses four transmembrane domains and two cytoplasmic termini [79]. LAPTM4B is regarded as an oncogenic protein as it is overexpressed in many human cancers [79,80] and correlates with poor prognosis [80,81]. LAPTM4B transforms normal human cells [82] and promotes the proliferation and migration of cancer cells [83]. LAPTM4B prolongs epidermal growth factor receptor (EGFR) signaling by inhibiting the intraluminal sorting and degradation of active EGFR, and this effect is counteracted by endosomal PIPKIγi5 and PI(4,5)P_2_ [84]. 

The degradation of activated EGFR is crucial for controlling EGFR signaling [85]. Upon activation by ligands, EGFR is rapidly internalized to endosomes, then sorted into intraluminal vesicles in the multi-vesicular endosomes or late endosomes [86,87], which fuse with lysosomes to enable EGFR degradation [88]. There are multiple EGFR agonists but EGF binds with high affinity and is not removed in the multivesicular body (MVB) [84]. The sorting of EGFR, an essential step for this regulation, is mediated by the endosomal sorting complex required for transport (ESCRT) [89,90,91]. Hrs, one of the ESCRT upstream subunits, contains ubiquitin-interacting motifs (UIM) that recognize ubiquitinated EGFR [90]. The ubiquitination of Hrs by the E3 ubiquitin ligases Nedd4-1 and Nedd4-2, on the other hand, blocks its UIM and ability to recognize ubiquitinated EGFR [92,93,94]. LAPTM4B promotes the ubiquitination of Hrs by recruiting Nedd4, interfering with the Hrs-dependent sorting of ubiquitinated EGFR [84]. PIPKIγi5 generates PI(4,5)P_2_, recruits SNX5, and directly binds LAPTM4B to inhibit its association with Hrs, thus preventing Hrs ubiquitination and enhancing EGFR intraluminal sorting and degradation [84] (Figure 2).

Inactive EGFR also acts as a signaling scaffold that is required for autophagy initiation [95]. In this pathway, inactive EGFR binds LAPTM4B upon serum starvation, resulting in its stabilization at endosomes. The EGFR-LAPTM4B complex recruits the exocyst subcomplex containing Sec5. The inactive EGFR-LAPTM4B-Sec5 subcomplex is required for basal and starvation-induced autophagy [95]. LAPTM4B and Sec5 promote the interaction of this EGFR complex with the autophagy inhibitor Rubicon, which in turn disassociates Beclin 1 from Rubicon to initiate autophagy. LAPTM4B regulates the role of inactive EGFR in autophagy via this molecular mechanism. Significantly, this pathway is positioned to control tumor metabolism and promote tumor cell survival upon serum deprivation or metabolic stress. LAPTM4B is positioned at the nexus between the inactive EGFR regulation of autophagy and active EGF-stimulated EGFR degradation and downregulation of EGFR receptor tyrosine kinase signaling.

### 3.7. IQGAP1 Scaffolds PI3,4,5P_3_ Synthesis at Membranes

The IQ-motif-containing GTPase-activating protein 1 (**IQGAP1**) is a multidomain protein responsible for scaffolding multiple signaling pathways [96,97]. The mitogen-activated protein kinase (**MAPK**) pathway is scaffolded by IQGAP1; Raf, MEK, and Erk, directly bind to IQGAP1 and are recruited in a sequential order of their phosphorylation in close proximity [98,99]. IQGAP1 also binds PIPKIγ in response to growth factors and the extracellular matrix (**ECM**) [100]. The phosphorylation of Ser1443 on IQGAP1 is required for its activation and binding to PIPKIγ via the IQ domain of IQGAP1 [100,101]. PI(4,5)P_2_ association with its Ras GTPase-activating protein (**GAP**) C-terminal (**RGCT**) domain, as well as the recruitment of Rac1 and Cdc42 via its GAP-related domain (**GRD**), allow the full opening of IQGAP1 [102,103,104], leading to de novo actin polymerization at the leading edge [105,106]. 

The activity of agonist-activated PI3K and the generation of PI3,4,5P_3_ were thought to occur at the plasma membrane [61,107]. However, the early literature suggested that PI3K was on intercellular membranes. Fractionation and imaging approaches indicated that agonist-stimulated PI3K activity was predominantly associated with intracellular endosomal membranes and to a lesser extent with the plasma membrane [108]. Moreover, various studies have reported PI3,4,5P_3_ generation to be either largely on the plasma membrane [42] or endomembranes [109], indicating that at least some agonist-stimulated PI3,4,5P_3_ may be intracellular. Consistent with this idea, the tumor suppressor PTEN, which dephosphorylates PI3,4,5P_3_ and inactivates PI3K/Akt signaling, is associated with endosomes in proximity to microtubules [110]. These data are puzzling because endosomal compartments appear to lack significant PI4,5P_2_ [61,107], the substrate for PI3K. However, the recent discovery that PI3,4,5P_3_ synthesis requires the IQGAP1 scaffold [111] may resolve this dilemma. IQGAP1 directly scaffolds the three sequential steps of PI3,4,5P_3_ synthesis. PI3,4,5P_3_ synthesis is initiated via PtdIns phosphorylation by PI4K to generate PI4P; PI4P is further phosphorylated by PIPKI to generate PI(4,5)P_2_; and finally, PI(4,5)P_2_ is phosphorylated by PI3K. All of these kinases are integrated into the IQGAP1 scaffold complex and the initiating substrate PtdIns appears channeled by the scaffold and sequentially phosphorylated to PtdIns(3,4,5)P_3_. As PtdIns is present on all cellular membranes, the IQGAP1 scaffold provides a molecular mechanism for PI3,4,5P_3_ synthesis in the absence of detectable PI(4,5)P_2_ on endosomal compartments. In addition, PDK1 and Akt, the downstream effectors of PI(3,4,5)P_3_, are also part of the IQGAP1 scaffold. In other studies, IQGAP1 was also found to interact with the mammalian receptor of rapamycin complex 1 (**mTORC1**), and this interaction sufficiently activated the full PI3K/Akt/mTOR pathway [112,113] (Figure 2). Specifically, IQGAP1 assembles PI4KIIIα, PIPKIα, and class I PI3K. The PI(4,5)P_2_-generating PIPKIα interacts with the IQ3 motif of the IQ domain on IQGAP1, while PI3K, responsible for phosphorylating PI(4,5)P_2_ to PI3,4,5P_3_, interacts with both the IQ and the WW domains of IQGAP1, spatially organizing PI3,4,5P_3_ production [111].

As PI3,4,5P_3_ is barely detectable in unstimulated cells, while PI(4,5)P_2_ is highly abundant in cells under basal and stimulated conditions, it has been postulated that stimulated PI3,4,5P_3_ synthesis requires the de novo synthesis of PI(4,5)P_2_ [114]. In IQGAP1, de novo PI(4,5)P_2_ produced by PIPKIα is channeled to PI3K for PI3,4,5P_3_ synthesis through the IQGAP1 complex, activating the Akt pathway [111]. The proximity and ratio of PIPKIα to PI3K in the complex are essential for the efficiency of PI3,45P_3_ production, emphasizing the importance of kinase scaffolding on IQGAP1 to facilitate the rapid and spatially regulated synthesis of PI3,4,5P_3_. These combined results support the de novo use of PtdIns in PI3,4,5P_3_ generation and this frees the localization of the PI3K signaling pathway from the limitation of existing pools of PI(4,5)P_2_ on specific membrane structures (Figure 3).

The discovery of the IQGAP1-PI3K pathway complex and the demonstration that the inhibition of IQGAP1-mediated PI3,4,5P_3_ production can selectively kill or block proliferation of cancer cells points to the therapeutic potential of targeting this complex [111]. Additional studies have demonstrated that the IQ3 motif of IQGAP1 controls its selectivity towards the PI3K-Akt pathway versus the ERK kinase pathway [115]. An IQ3 motif peptide inhibited Akt activation, cell proliferation, invasion, and migration independent of the IQGAP1-scaffolded Ras-ERK pathway [115]. These findings suggest novel therapeutic strategies to specifically target the IQGAP-dependent scaffolding of PI3k/Akt signaling without blocking the ERK pathway.

### 3.8. MAP4 Scaffolds the PI3K-Akt Pathway Predominantly on Endosomal Membranes

The PI3K-Akt pathway was assumed to co-localize at the plasma membrane with the activating receptor tyrosine kinases, such as EGFR or the insulin receptor, where the PI3K substrate PI(4,5)P_2_ is concentrated [96,97]. However, there is little evidence for this subcellular localization of PI3K or its spatial recruitment to receptor tyrosine kinase to support this established belief. Indeed, multiple studies have shown that PI3K co-localizes with endosomes [110], and PI3,4,5P_3_ generation and Akt activation occurs on endomembranes stimulated by ligand-activated receptors [109,116,117]. Furthermore, the assembly of the IQGAP1-PI3K complex provides a mechanism for the de novo generation of PI3,4,5P_3_ from PI [111]. However, the mechanism(s) by which the PI3K pathway is targeted to endosomes remain(s) unknown. 

The interaction of PI3K with microtubule-associated protein 4 (**MAP4**) appears to be one such mechanism. MAP4 is a major non-neuronal microtubule-associated protein that binds and controls the localization of the vesicle-associated PI3Kα with endosomes [118]. Rather than initiating the PI3K-Akt pathway on the plasma membrane, activated receptor tyrosine kinases, specifically EGFR, are endocytosed and sorted to the endosomal compartments where PI3,4,5P_3_ synthesis through PI3Kα and subsequent Akt activation occur [118]. Additionally, MAP4 was shown to be required for the association of PI3Kα with activated receptor tyrosine kinase [118]. Moreover, knocking down MAP4 impaired agonist-stimulated PI3,4,5P_3_ generation and Akt phosphorylation, resulting in the inhibition of cell proliferation and invasion [118]. Overall, these findings underscore the critical role of MAP4as an essential component of the PI3K-Akt signaling pathway (Figure 2).

MAP4 was initially discovered in a mass spectrometric analysis of PI3Kα isolated via immunoprecipitation [118]. An interaction between PI3K and MAP4 indicated PI3,4,5P_3_ generation and Akt activation at intracellular membrane vesicles as these are adjacent to microtubules [109,116,117]. The observed interaction of PI3Kα with MAP4 and PI3Kα vesicle co-localization with activated EGFR receptors at internal vesicles indicated that PI3,4,5P_3_ generation and Akt activation occur on endomembranes. Within minutes of binding agonists, receptor tyrosine kinases are endocytosed and the active receptor is sorted into the endosomal compartments where it remains active for 30 to 60 min [119,120,121]. This indicates that the majority of receptor signaling takes place at endosomes and PI3,4,5P_3_ synthesis and Akt activation largely occur after the receptors are internalized. Additionally, these data suggest that endomembrane compartments are the predominant site of PI3,4,5P_3_ generation and Akt activation downstream of activated receptor tyrosine kinases.

Immunofluorescent time course analyses of PI3,4,5P_3_ and phosphorylated Akt following EGF stimulation revealed that they co-localize with microtubules and the endosome compartments [118]. This is consistent with prior reports indicating that the activation of Akt requires the continual engagement of the Akt PH domain with PI3,4,5P_3_ or PI(3,4)P_2_ [122]. Analyses of the interaction of PI3Kα with MAP4 demonstrated that MAP4 has a preference for the p110α catalytic complex containing the p85α adaptor subunit, and the C2 domain of p110α directly binds to MAP4 through the repeats of the microtubule-binding domain (**MTBD**) [118]. The MAP4-PI3K interaction was regulated via agonist stimulation and colocalized with early endosomal markers [118]. Moreover, the MAP4-PI3K complex was associated with IQGAP1 using a proximity ligation assay (**PLA**) and the MAP4-IQGAP1 interaction was stimulated via EGF. Notably, these complexes all co-localized at endosomes with EGFR. These interactions define a mechanism by which agonist-stimulated PI3,4,5P_3_ generation and Akt activation occur at the endosome [111,118].

Not only does this study demonstrate a model of agonist-stimulated PI3K-Akt on endosomes that is supported by extensive data, but it also highlights the potential of the MAP4-PI3K interaction as a therapeutic target as PI3K hyperactivation occurs frequently in many cancers and MAP4 is overexpressed in diverse tumors [123,124]. Significantly, MAP4 is also required for a dominant active PI3K mutant to activate Akt [118]. MAP4 knockdown impairs Akt activation in cell lines with diverse PI3K-Akt pathway mutations, including dominant active PI3K, PTEN null, K-Ras activation, and PI3K overexpression, and MAP4 loss inhibits the proliferation and invasion of these cell lines [118].

## 4. PI Scaffolding Complexes in Autophagy

Autophagy is the “self-eating” of cytosolic components and is an essential cellular homeostasis mechanism under conditions of nutrient scarcity. It is a multi-step process, forming specific autophagic organelles in each step [125]. Organelles initiate from the membrane formation of omegasomes, gradually elongate to phagophores and enclose to generate an autophagosome, and then fuse with lysosome to form autolysosomes, which degrade the cargo. Phosphoinositide metabolism plays a key role in every step of autophagy. Different phosphoinositides recruit specific compartments on membrane structures, mediating the proper generation and activation of autophagic organelles [125,126] (Figure 4).

### 4.1. PI Complexes Regulate Autophagosome Maturation

The initiation of autophagic membrane structures is carried out by the protein Beclin1 [127,128,129]. Autophagy is stimulated by multiple stressors including serum starvation [130]. Upon serum starvation, inactive EGFR internalizes into LAPTM4B-positive endosomes [95]. Inactive EGFR and LAPTM4B stabilize each other at the late endosome and interact with multiple subunits of the exocyst complex, including Sec5. The EGFR-LAPTM4B-Sec5 complex recruits the autophagy inhibitor Rubicon, which in turn disassociates Rubicon from the Beclin1 complex and initiates the autophagy process [95]. The inactive EGFR-LAPTM4B complex specifically regulates autophagy, whereas the active EGF-EGFR complex prolongs EGFR signaling by blocking internalization into the lysosome where it is downregulated via the degradation of the receptor. 

Beclin1 initiates the formation of the Beclin1-Vps34-Vps15 scaffolding complex, and the complex is recruited to the ER by ATG14 for autophagy specificity [131,132], while Vps34 generates PI(3)P from PtdIns. Vps34, and together with Vps15, Vps30, ATG14, and ATG38, forms PI3K complex I (PI3KCI) [131,133,134,135]. In yeast, PI3KCI constitutively binds to the vacuolar membrane protein Vac8 through ATG14 [136]. Upon autophagy initiation, ATG 1 recruits ATG9, ATG13, and ATG17-ATG31-ATG29 to assemble the ATG1 complex [137,138,139]. In an ATG1 kinase activity-dependent manner, PI3KCI associates with the ATG1 complex via the ATG38-ATG1 complex and Vps30-ATG9 interactions, anchoring PI3KC1 to the pre-autophagosomal structure for PI(3)P production [136]. A minor pool of PI(3)P is also generated from PI3KC2 in starvation-induced autophagy [22]. DFCP1, a protein containing the PI(3)P-binding FYVE domain, is recruited to these pools of PI(3)P, and together regulate the biogenesis of lipid droplets [140], acting as markers of the omegasomes [141].

PI(3)P also plays a critical role in the maturation of phagophores to autophagosomes. WIPI-2 binds to PI(3)P on the omegasomes and mediates ATG16L-ATG5-ATG12 complex formation [142]. The ATG16L complex acts together with ATG3 as an E3-like ligase and E2-enzyme for LC3 lipidation, coupling LC3 with phosphatidylethanolamine (**PE**) to generate LC3 II, which regulates the closure, fusion, and transport of the autophagosome [143,144]. Another PI(3)P effector, WIPI-4, forms a complex with ATG2 and ATG9 on the PI(3)P pools, tethering the autophagic membrane to the ER for lipid transfer to elongate this membrane structure [145].

Phosphoinositides are also involved in the initiation of autophagosomes. ATG9A, the ubiquitously expressed variant of ATG9, locates to autophagosomes on vesicle compartments where it binds to Arfaptin2 and PI4KIIIβ, bringing the complex to autophagic sites, and promoting the generation of PI(4)P [146]. PI(4)P then recruits the ULK-ATG13 complex, which activates Vps34 [146,147], enhancing autophagy as positive feedback. PI(5)P synthesized by PIKfyve on phagophores can act as a binding partner of DFCP1 and WIPI2 independent of the PI(3)P pool, forming scaffolding complexes and mediating the biogenesis of autophagy as mentioned above [148]. PI(4,5)P_2_ plays a more controversial role in autophagy initiation. On the one hand, PI(4,5)P_2_ phosphorylated from PI(5)P by PIPKIIs negatively regulates autophagosome biogenesis by depleting the PI(5)P pool. On the other hand, PI(4,5)P_2_ contributes to the autophagic membrane generated from endosomes and the ER. On ATG16L1-positive endosomes, SNX16, a PX-domain-containing protein, that specifically interacts with PI(4,5)P_2_, ATG16L1, and LC3, mediates the delivery of the ATG16L1 complex to the autophagic site and the lipidation of LC3 [149], while another pool of PI(4,5)P2 generated by PIPKIγi5 on the ER binds to ATG14, stabilizing the autophagy-specific Beclin-Vps34-Vps15-ATG14 scaffolding complex [150]. 

### 4.2. PI Complexes Regulate Autolysosome Fusion

After the maturation of the autophagosome, the next step of cellular content degradation is fusion with lysosomes, where phosphoinositide scaffolding complexes also play a vital role. To fuse with lysosomes, the autophagosome must first be transported by microtubules, and the tethering of the autophagosome and microtubule depends on FYCO1-Rab7-PI(3)P-LC3 II complex formation [151,152]. This allows the autophagosome to be transported to the cell periphery, where the lysosomes reside. Upon coming into proximity with the lysosomes, lysosome membrane protein TECPR1 interacts with autophagosome membrane proteins ATG5–ATG12, enabling its binding to autophagosome PI(3)P through its PH domain. This intervesicular complex facilitates lysosome–autophagosome fusion [153]. 

PI(4)P presented on both the autophagosome and lysosome membranes is crucial for their fusion. Gamma-aminobutyric acid receptor-associated protein (**GABARAP**) binds and recruits PI4KIIα to the autophagosome for PI(4)P generation. Knocking down this complex results in the enlargement of the autophagosome and the accumulation of IC3 II, indications of impaired autophagosome–lysosome fusion [154]. On the lysosomes, PI(4)P synthesis is controlled by PI4KIIIβ, and this PI(4)P pool interacts with Rab7, anchoring the PH domain-containing protein family member 1 (**PLEKHM1**) and the homotypic fusion and protein-sorting (**HOPS**) complex, and PLEKHM1 binds to the LC3/GABARAP complex via the LC3 interaction region [155]. In this way, the PI(4)P pool on both vesicles regulate the tethering and fusion of autophagosomes and lysosomes. Other studies indicate that PI(4,5)P_2_ also controls this process via multiple mechanisms. The conversion of PI(4)P into PI(4,5)P_2_ by PIPKIγ inactivates Rab7 and releases PLEKHM1 from endosomes, allowing the complex’s formation at autolysosome fusion sites [156], while ATG14, a PI(4,5)P_2_ effector, binds to the soluble *N*-ethylmaleimide-sensitive factor attachment protein receptor (**SNARE**) domain of STX17 and SNAP29, stabilizing the STX17-SNAP29 binary t-SNARE complex on autophagosomes to facilitate the fusion [157].

### 4.3. PI Complexes Regulate Autophagic Lysosome Reformation

After degradation, autolysosomes are required to be recycled back into new lysosomes. If nutrients are scarce, this is achieved via a process called autophagic lysosome reformation (**ALR**). Studies show that a specific pool of PI(4,5)P_2_ mediates this process. PtdIns on the autolysosome is first phosphorylated by PI4KIIIβ to PI(4)P, to be used as a substrate for PIPKIα/β to generate PI(4,5)P_2_ [158,159]. The PI(4,5)P_2_ on the autolysosome surface recruits the adaptor proteins AP2 and AP4, and clathrin to induce membrane budding and tubule formation [159]. Then, the motor protein KIF5B interacts with PI(4,5)P_2_ in a clathrin-dependent manner, driving autolysosome tubulation and ALR [160].

## 5. The PI Scaffolding Complex Regulates Rapid Lysosome Repair

Lysosomes are membrane-confined organelles containing an array of digestive enzymes, capable of breaking down macromolecules and lipids. Lysosomes are involved in endocytic and autophagic degradation, as well as nutrient sensing, growth signaling, the stress response, and immunity. As lysosome activity requires an acidic luminal pH, which would be toxic to the cell, the integrity of lysosomes is crucial for proper lysosomal activity and cell survival [161]. A hallmark of lysosomal dysfunction, lysosomal membrane permeabilization (**LMP**) can be triggered by stress and lead to lysosomal-leakage-induced cell death. Old and injured lysosomes with severe LMP are selectively degraded via macroautophagy, known as lysophagy [162,163]. Less severe LMP can be reversed through rapid membrane repair pathways, including the endosomal sorting complex required for transport (**ESCRT**) [164] and the phosphoinositide-initiated-membrane-tethering and lipid transport (**PITT**) pathway [165].

In the PITT pathway, PI4KIIα is recruited to damaged lysosomes, possibly through lysosomal Ca^2+^ release, and on-site PI(4)P synthesis. PI(4)P in turn recruits its effectors in the oxysterol-binding-protein (**OSBP**)-related protein (**ORP**) family, namely ORP9, ORP10, ORP11, and OSBP, which bind PI(4)P through their respective PH domains. ORP9-ORP10 heterodimers, ORP9-ORP11 heterodimers, and OSBP homodimers directly bind ER proteins through their FFAT motif, mediating membrane tethering and establishing MCS between the damaged lysosomes and the ER network (Figure 2). The ORP dimers also function as lipid transfer complexes at MCS, exchanging PI(4)P for phosphatidylserine (**PS**). The PS then recruits ATG2, which rapidly repairs the damaged lysosomes through its lipid transport activity [165]. 

## 6. Conclusions

It is now clear that scaffolding complexes are widely regulated by phosphoinositides and in turn regulate phosphoinositide signaling, which is involved in the biogenesis and interconversion of phosphoinositides in cytoplamic second messenger signaling, membrane sorting, autophagy, and lysosome repair. Many of the scaffolding complexes described in this review either recruit kinases and phosphatases to regulate phosphoinositide generation or recruit downstream effectors of the phosphoinositides to enhance phosphoinositide signaling. Scaffolding proteins such as IQGAP1 serve as a physical platform on which both kinases and effectors assemble together to enable the highly dynamic spatial and temporal regulation of the PI3K-Akt pathway. Phosphoinositide scaffolding occurs on the plasma membrane, cytosolic vesicles, including the Golgi, endosomes, the ER, autophagosomes and lysosomes. The breadth of the functions and localizations of the phosphoinositide scaffolding complex emphasize the crucial role of phosphoinositide generation in cytoplamic signaling and membrane trafficking, enabling the specificity and efficiency that is brought about by scaffolding functional components. Although this review has focused on cytoplasmic scaffolds that control phosphoinositide regulation, emerging evidence points to the critical functional roles of phosphoinositide scaffolding in the nucleus [166,167,168,169].

## Figures and Tables

**Figure 1 biomolecules-13-01297-f001:**
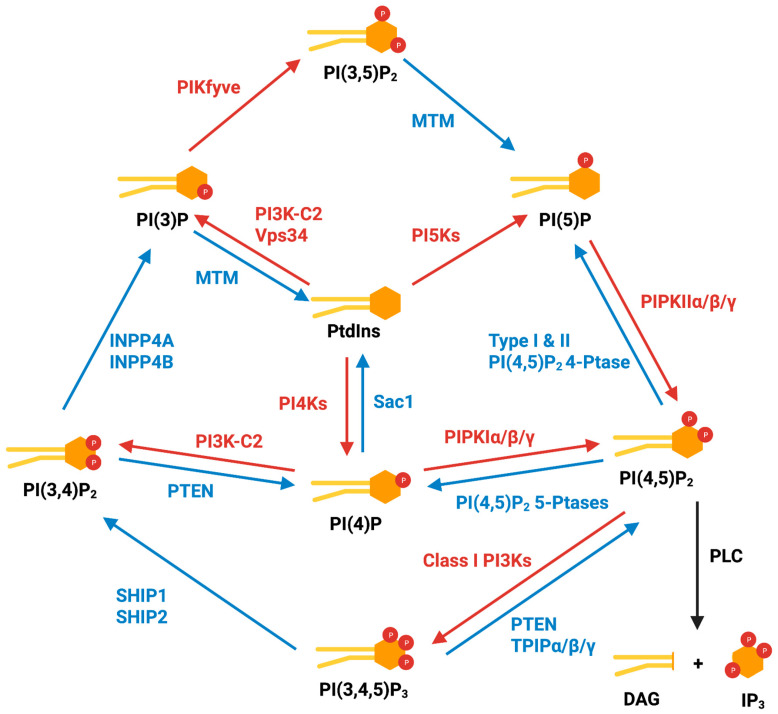
Overview of cytoplasmic PI metabolism. The diagram shows an overview of PI family members and their metabolic regulation by PI-modifying enzymes. Red lines and enzyme names indicate phosphorylation pathways, while blue lines and enzyme names indicate dephosphorylation pathways. Figure created with biorender.com.

**Figure 2 biomolecules-13-01297-f002:**
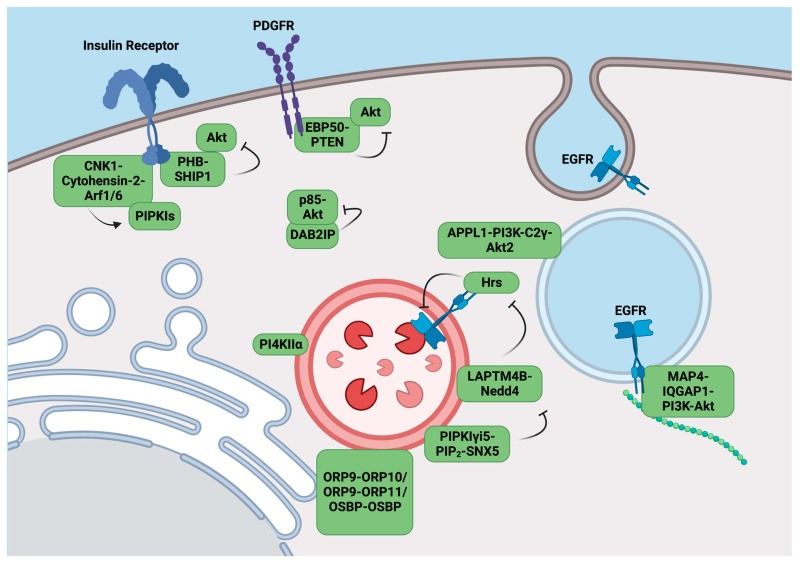
Overview of the cytoplasmic scaffolding of the PI3K/Akt pathway. The PI3K/Akt pathway is initiated by multiple cell surface receptors and facilitated by different scaffolding proteins. CNK1 is recruited to the plasma membrane via insulin stimulation, bringing cytohensin-2 with it to activate PIPKI-mediated PI(4,5)P_2_ synthesis. EBP50 and PHB are both inhibitory scaffold proteins that promote PI(3,4,5)P_3_ dephosphorylation by recruiting PTEN and SHIP1, respectively, to sites of Akt activation. DAB2IP also functions as an inhibitor of the PI3K/Akt pathway by scaffolding and translocating p85 and Akt from the membranes to cytosol. APPL1, IQGAP1, and MAP4 are endosomal scaffolding protein for Akt activation. APPL1 scaffolds PI3K-C2γ and Akt2 in response to multiple receptor stimulations, and promotes Akt2 activation via PI(3,4)P_2_ synthesis. LAPTM4B prolongs EGFR signaling via Hrs ubiquitination, resulting in the inhibition of EGFR sorting, while the PIPKIγi5-PI(4,5)P2-SNX5 complex on the endosome counteracts this effect. IQGAP1 and MAP4 form a PI(3,4,5)P_3_ synthesis complex by recruiting PI4KIIIα, PIPKIα, and PI3Kα in close proximity, and channel de novo PI(4)P and PI(4,5)P_2_ to PI(3,4,5)P_3_ synthesis under EGF or insulin stimulation. IQGAP1 also interacts with the PI(3,4,5)P_3_ effectors PDK1 and Akt, scaffolding the entire PI3K/Akt pathway in a processive manner. Additionally, illustrated is the PITT pathway of rapid lysosome repair, in which PI(4)P generated by PI4KIIα recruits ORP dimers as lipid transfer complexes. Figure created with biorender.com.

**Figure 3 biomolecules-13-01297-f003:**
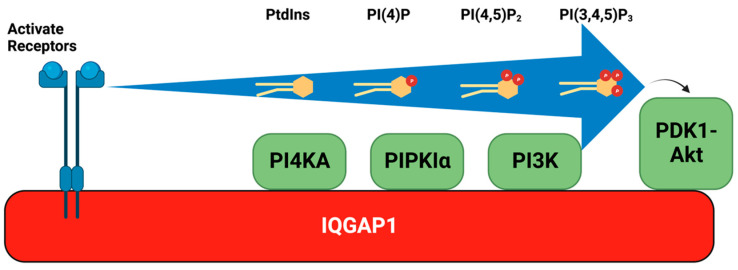
Processive PI3K/Akt pathway assembled on IQGAP1. This model demonstrates how PI(3,4,5)P_3_ synthesis is scaffolded by the IQGAP1 scaffold complex. Kinases phosphorylating different sites on the inositol head of PtdIns are assembled on IQGAP1 in close proximity, catalyzing the synthesis of PI(3,4,5)P_3_ in a step-by-step manner. By utilizing PtdIns rather than the existing PI(4,5)P_2_ pool, the IQGAP1 scaffold complex enables rapid PI(3,4,5)P_3_ synthesis and Akt activation in response to stimulation on endosomes, which are typically devoid of native PI(4,5)P_2_. Figure created with biorender.com.

**Figure 4 biomolecules-13-01297-f004:**
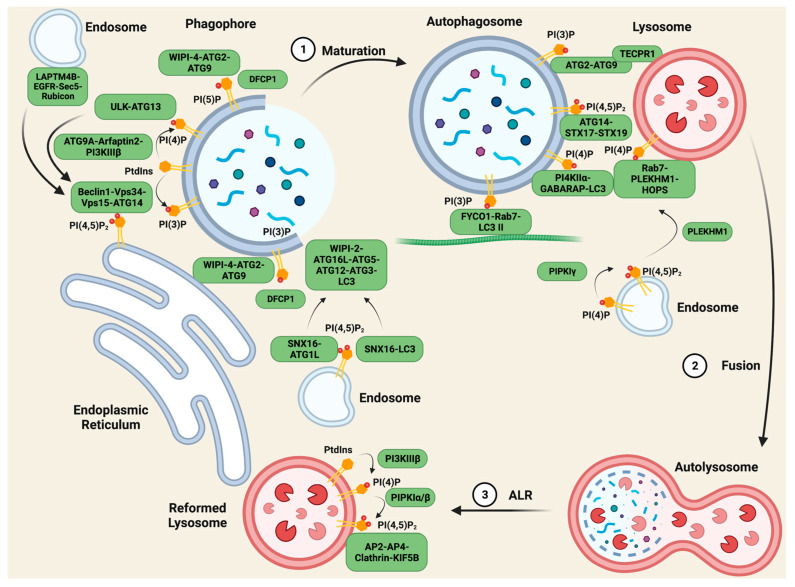
PIs regulate scaffolding complexes in autophagy. During the maturation of autophagosomes, the fusion between autophagosomes and lysosomes, and the reformation of lysosomes after degradation, membrane PIs play vital roles in recruiting specific scaffolding complexes to mediate lipid transfer, vesicle transport, and membrane tethering. The most important PI species in autophagy are PI(3)P and PI(4)P, as they anchor multiple crucial complexes in different steps of autophagy, while PI(5)P and PI(4,5)P_2_ also have unique roles. Figure created with biorender.com.

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
