# Peer review of "Regulation of Phosphoinositide Signaling by Scaffolds at Cytoplasmic Membranes"

_biomolecules, 2023, doi:10.3390/biom13091297_

Round 1

Reviewer 1 Report

This is a comprehensive and well-written review focusing on the role of scaffold proteins in the regulation of the PI3K/Akt pathway. As far as I know, the main literature in the field is cited.

I found some minor mistakes in the text:

- line 19 (abstract): PI3K/Akt

- line 36: array

- line 120: stimuli

- line 131: “Scaffolding complexes target the PI3K/Akt pathways to specific compartments” is better.

- line 189: … is a member of the Ras-GAP (GTPase-activating protein) family.

- line 308: there is little evidence..

- line 315: remain

- line 342: indicating

- line 401: a critical role in…

Author Response

Thank you very much for the review and corrections. We have edited our manuscript accordingly.

Reviewer 2 Report

In their review manuscript entitled "Regulation of Phosphoinositide Signaling by Scaffolds at Cytoplasmic Membranes" Tianmu Wen et al discuss the role of scaffold proteins in organizing phosphoinositide (PI) metabolism and signaling. They focus primarily on PI3K/Akt signaling but they also discuss scaffolding complexes to membrane structures that coordinate PI-dependnt vesicle formation, fusion, and reformation during autophagy.  The authors highlight their recent high-profile work on MAP4 and IQGAP1 scaffolds in the context of endomembrane PI signaling. The manuscript is well-curated and organized with very informative and elegant figures to illustrate the modes of PI regulation and their functions.

I have only four comments and a couple of minor corrections

Comment 1: The authors refer to cytoplasmic or cytosolic PIs in several sections of the manuscript. Examples are cytoplasmic PIs (line 11, abstract), section 2 title, Figure 1 title, PIP3 in the cytosol (line 111), etc... Although I understand the reasoning for using this "terminology" it strikes me as a bit awkward. Furthermore, for the the non-specialist, this terminology may be also misleading, as it would suggest two pools of PIs a cytosolic and a membranous. My suggestion for the authors is to refrain from using the term cytosolic/cytoplasmic for PIs throughtout the text (reduce it substantially).

 Comment 2: The authors provide a figure for the basic metabolism of PIs (kinases/phosphatases). It would be helpful and relevant if the authors here mention whether some reactions are observed only in vitro with purified enzymes and have not been verified in cells. For example, has direct  phosphorylation of PI to PI5P been verified in cells/in vivo? These seems to be no specific  reference concerning this.

Comment 3: line 256 and statement on "PIP3 generated largely on endomembranes". In my view, this statement needs to be re-evaluated. The Sato et al Nat Cell BIol paper describes PIP3 on endomembranes using overexpressed fluorescent FRET-based sensors. Using other techniques there has been ample evidence for production of PIP3 on the plasma membrane.  For example, a more detailed analysis by Liu et al published in Molecular Cell in 2016 showed that agonist-induced PIP3 is primarily generated on the PM while PI34P2 is generated on both PM and endomembranes (https://doi.org/10.1016/j.molcel.2018.07.035). Thus, in my opinion, the authors should provide a more focused paragraph on PIP3/PI34P2 production on the PM and endosomal membranes. At minimum, this sentence needs to be rephrased and re-referenced to point that PIP3 is also produced on the plasma membrane.

Comment 4. Section 3.5. APPL1 scaffolds Akt2 on endosomes for PI(3,4)P2 activation.  Here an introduction of how PI34P2 is formed on endosomes might provide the context. For example, is it via a PI3K class II or a PI3K class I-SHIP2 mode?

Minor corrections

1. Abstract, line 19: correct PI3K?Akt signaling

2. line 411-412, "... it binds to Arfaptin2 and PI3KIIIβ, bringing the complex to autophagic sites, and promoting the generation of PI(4)P". The kinase responsible should probably be PI4KIIIβ not PI3KIIIβ, no?

Author Response

Thank you very much for the review and the comments. These are important points and we revised the manuscript to address them.

For Comment 1, we changed most of the word “cytosolic” to “cytoplasmic”, only using the term “cytosol” when we are referring to non-membrane structures to avoid confusion.

For Comment 2, we cited more research papers and reviews that indicate the cellular activation of the kinases and phosphatase, and mentioned the absence of evidence of cellular activity in the case of PI(3,4)P2synthesized by PI3K from PI(4)P.

For Comment 3, we recognized the research showing the production of PI(3,4,5)P3 on the plasma membrane by rephrasing the sentence and citing the corresponding work.

For Comment 4, we specified the generation pathway of PI(3,4)P2 on endosomes by SHIP2.

For the minor corrections, we edited the manuscript accordingly.